# Identification of novel vertebral development factors through UK Biobank candidate gene search and body imaging analysis

Zhuopin Sun[1,2,4], Jiru Han[1,2,4], Liam G. Fearnley[1,2], Edwina McGlinn [3] & Melanie Bahlo [1,2] ✉

Numerical variations and transitional anatomy in the human vertebral column represent a significant yet understudied aspect of skeletal development with potential effects on multiple physiological systems. Utilising UK Biobank data, we integrated genetic analysis with deep learning-based multimodal body imaging to investigate genetic factors associated with thoracic and lumbar spine anatomy. We identified three key genes, *GPC3*, *NR6A1*, and *VRTN*, associated with numerical variations of the lumbar vertebrae and ribs, with *VRTN* reported for the first time in humans as influencing vertebral development. Our findings reveal significant associations between these genetic variants, vertebral and rib anomalies, and increased prevalence of chronic pain. This study highlights the genetic underpinnings of vertebral development and demonstrates the utility of combining imaging and genetic data to uncover skeletal variation and its health implications for population health.

The evolution of the human axial skeleton resulted from a series of regionalised, anatomical characteristics specialised for supportive, locomotive, protective, and other essential functions[1,2]. The human vertebral column typically consists of 33 vertebrae: 7 cervical, 12 thoracic, 5 lumbar, 5 sacral, and 4 coccygeal, along with the skull, ribs, and sternum. However, numerous clinical and anthropologic studies have reported the presence of anatomical variation, such as transitional vertebrae, in 3–36% of the population, along with rib variations[3–7]. In addition, true numeric changes are rarer, with lumbar ribs observed in roughly 1% of individuals[8]. These variations can affect overall health, influencing the nervous, cardiovascular, neurological, urinary, and reproductive systems[9].

The development and evolutionary changes in the vertebral column have been found to be controlled by regulatory genes and growth factors that act through specific inductive mechanisms. Notably, the *Hox* genes play a crucial role in the regulation of vertebral segments[10–12]. Additionally, several genes identified in mice studies can influence *Hox* gene expression, leading to changes in the vertebral column[13–15]. For example, the Growth and Differentiation Factor 11 (*GDF11*) has been identified as a key molecule that determines positional identity of the axial skeleton by modulating *Hox* gene expression[15,16]. Furthermore, the nuclear receptor subfamily 6 group A member 1 (*NR6A1*) has also been found to control *Hox* expression dynamics and serves as a master regulator of vertebrate trunk development[13]. Recent studies have identified *NR6A1* variants associated with numerical variations in the lumbar vertebrae and ribs[17,18]. In humans, abnormal development of the skeletal system is often associated with Mendelian disorders[19]. For example, six lumbar vertebrae have been observed in Simpson-Golabi-Behmel syndrome, a rare disease linked to mutations in the Glypican-3 (*GPC3*) and Glypican-4 (*GPC4*)[20,21]. Recently, the presence of four lumbar vertebrae has been identified in humans with oculo-vertebral-renal (OVR) syndrome and congenital renal, vertebral, and uterine anomalies, linked to *NR6A1* mutations[17,18]. Genetic variation in *GDF11* was also found to be associated with vertebral hypersegmentation and orofacial anomalies[22].

Despite valuable insights from molecular biology and mouse models about the genetic factors regulating vertebral column development, most known variants are extremely rare (Minor allele frequency (MAF) < 0.00001 or MAF = 0 in gnomAD) and are typically associated with severe phenotypic effects[23]. A recent genome-wide association (GWAS) study, using the UK Biobank (UKB), has provided evidence of common genetic factors that influence skeletal proportions for lower and upper limbs, revealing the complex relationships between genetics and the structure of the skeleton[24]. However, there has been limited research focusing on population datasets to explore the genetic determinants of vertebral anatomical variations, which has controversial clinical significance. Furthermore, it is difficult to characterise, especially at the scale required for population-based studies[25].

[1]Genetics and Gene Regulation, The Walter and Eliza Hall Institute of Medical Research, Parkville, VIC, Australia. [2]Department of Medical Biology, The University of Melbourne, Parkville, VIC, Australia. [3]Australian Regenerative Medicine Institute, Monash University, Clayton, VIC, Australia. [4]These authors contributed equally: Zhuopin Sun, Jiru Han. ✉e-mail: bahlo@wehi.edu.au

**Table 1 | Candidate gene and variant summary**

| Gene symbol | SNP | Mutation type | AZPheWAS Association | | MAF | phyloP | CADD | ClinVar | OMIM |
|---|---|---|---|---|---|---|---|---|---|
| | | | 20015#Sitting height | Other Associations (P < 1e -5) | | | | | |
| | | | BETA  P | | | | | | |
| VRTN | 14-74357184-G-A rs138257884 c.401G > A p.Arg134His | Missense | −0.56  4.98e- 06 | No associations observed | 0.003 | 8.77 | 26 | Not Reported | Not Reported |
| NR6A1 | 9-124524718-T-C rs752867091 c.1351+3A > G | Splice region | −0.32  5.97e- 06 | 51#Seated height | 0.01 | 6.36 | 22.2 | Not Reported | Not Reported |
| ACVR2B | 3-38483237-C-T rs144370188 c.1444C> T p.Arg482Trp | Missense | −0.34  2.83e-07 | 50#Standing height | 0.03 | 4.74 | 26.1 | Benign/ Likely benign | Heterotaxy, visceral, 4, autosomal (613751) |
| GPC3 | X-133692376-C-T rs11539789 c.1285G > A p.Val429Met | Missense | 0.06  3.02e- 08 | 50#Standing height 51#Seated height 20150#Forced expiratory volume in 1- second (FEV1) Best measure 20151#Forced vital capacity (FVC) Best measure 3062#Forced vital capacity (FVC) 3063#Forced expiratory volume in 1- second (FEV1) | 0.46 | 7.74 | 23.2 | Benign/ Likely benign | Simpson-Golabi- Behmel syndrome, type 1 (312870) Wilms tumor, somatic (194070) |

Positions refer to GRCh38 reference genome. BETA represents the SNP effect size on "20015#Sitting height" as reported in the AstraZeneca PheWAS browser, while P indicates the p value. MAF (Minor Allele Frequency) is shown as a percentage. PhyloP denotes conservation scores, with higher scores indicating greater conservation. CADD (Combined Annotation Dependent Depletion) scores assess predicted deleteriousness. MAF, PhyloP, and CADD values were obtained from gnomAD v4.1.

The gold standard for assessing vertebral morphology is Computed Tomography (CT) scans. When using X-rays or spinal Magnetic Resonance (MR) images for assessment, it is advised to employ multiple imaging modalities and exercise caution to distinguish artifacts and anatomical variations from true pathology[26]. The UKB only has Dual-energy X-ray Absorptiometry (DXA) and body Dixon MRI scans, but has the merit of a large sample size with sufficient power for common variant studies. The UKB has proven to be a very powerful resource for imaging-based studies utilising the optical coherence tomography[27], DXA[28–30], and body MRI[31] to identify new genetic drivers of human physiology.

In the present study, we employed both top-down and bottom-up approaches to identify novel, but rare genetic variants in three genes (GPC3, NR6A1, and VRTN) associated with numerical variation of the lumbar vertebrae and ribs. Utilising deep learning-based methods to integrate multi-modal body imaging data from the UKB, we achieved detailed phenotyping of vertebral and rib anatomy in the thoracic and lumbar spine. We focused on the more severe forms of vertebral abnormalities as they are more likely to present as congenital anomalies with visible changes on current imaging modalities. We further demonstrated that these anatomical variations are associated with an increased prevalence of pain, suggesting the clinical relevance of our findings, and providing new insights into the genetic influences underlying these traits. This study also demonstrates that UKB data can be utilised to investigate vertebral traits in the future.

## Results
### Candidate genetic variants selection
We identified a gene panel comprised of 51 genes that related to numeric variations of the thoracic and lumbar vertebrae from Human Phenotype Ontology (HPO)[32] and the Mouse Genome Informatics (MGI) phenotype databases[33]. Examination of genetic variation in our gene panel identified four genetic variants in four genes potentially contributing to vertebral number variation: rs138257884 in VRTN, rs752867091 in NR6A1, rs144370188 in ACVR2B, and rs11539789 in GPC3. These variants were selected based on their associations with either increased or decreased sitting

height, a potential surrogate marker for the presence of additional or fewer vertebrae (Table 1).

NR6A1 has recently been found to be associated with numerical variations in the lumbar vertebrae and ribs in ultra-rare variant clinical studies[17,18]. The VRTN gene has previously been linked to altered vertebral number in animal models[34,35], but no human gene-phenotype relationships or associated diseases are documented in the HPO or Online Mendelian Inheritance in Man (OMIM) databases[36]. Variants rs138257884 (VRTN) and rs752867091 (NR6A1) are not associated with any known GWAS findings and lack clinical significance in ClinVar[37]. These variants had high phyloP conservation scores (base-wise conservation score across 241 placental mammals, gnomAD v4.1), with scores of 8.77 for rs138257884 and 6.36 for rs752867091, indicating strong evolutionary constraint and suggesting potential functional significance.

ACVR2B has been linked to increased thoracic vertebrae in mouse (MGI) and is implicated in various visceral heterotaxy syndromes in humans (HPO and OMIM). The variant rs144370188 is classified as benign or likely benign for heterotaxy, visceral, 4, autosomal (OMIM: 613751), but may still indirectly influence vertebral numeric variation. Additionally, ACVR2B mediates GDF11 signalling in axial vertebral patterning[38], with GDF11 identified as a key molecule in determining the positional identity of the axial skeleton by regulating Hox gene expression[15,16].

GPC3 has been identified as the major gene causing Simpson-Golabi-Behmel syndrome (OMIM:312870), a rare overgrowth syndrome with multiple congenital anomalies. Previous studies have shown that deletion of exon 6 in GPC3 can result in six lumbar vertebrae[39]. However, the variant rs11539789 identified in this study is classified as benign or likely benign for other conditions, such as Wilms tumour, and has no documented associations with vertebral phenotypes or GWAS findings.

### Association of variant carrier status with physical measures
We explored whether selected candidate variants, potentially linked to vertebral numeric variation, are associated with physical measures, such as body size measurements and pulmonary function indicators, as surrogates

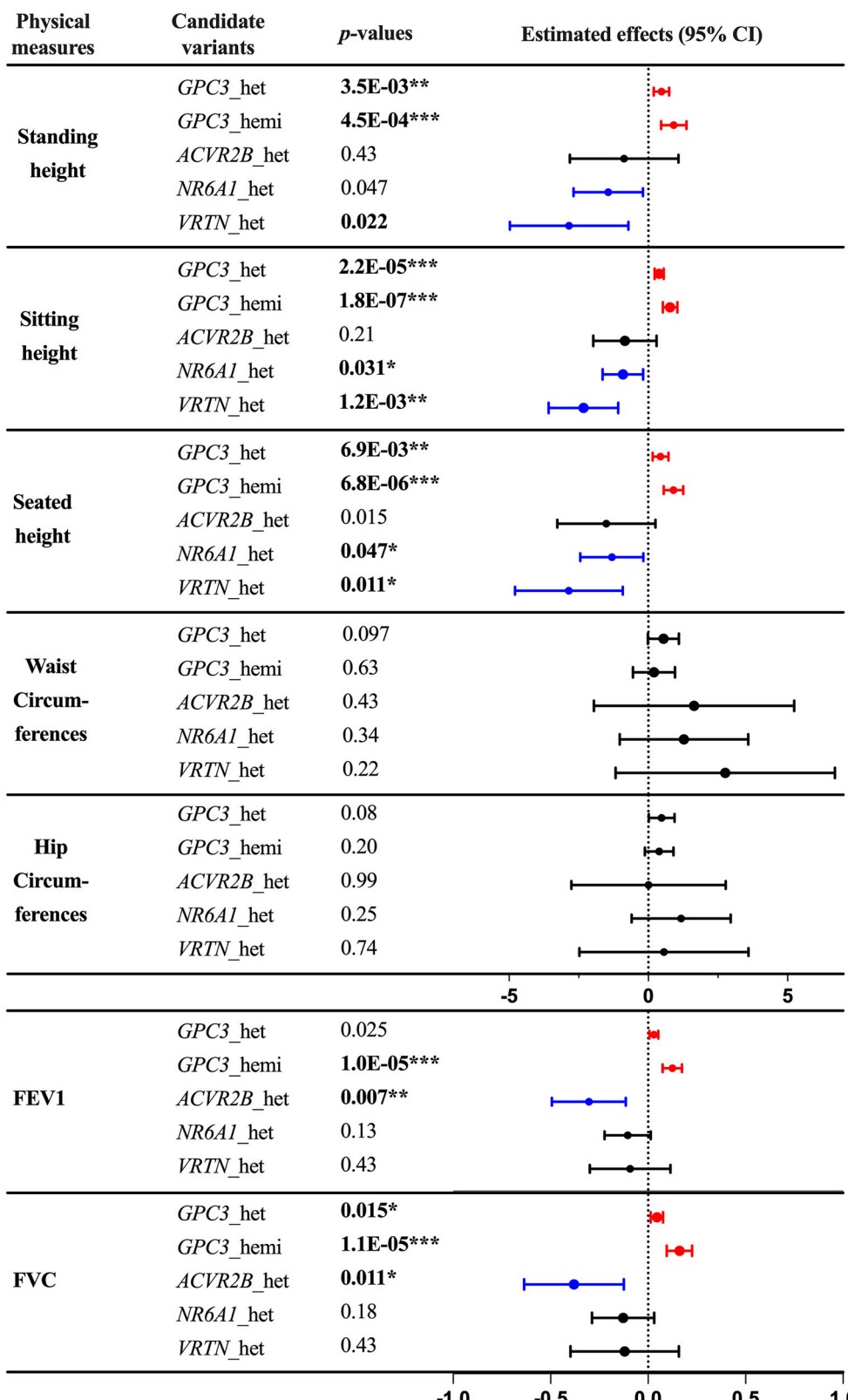

**Fig. 1 | Summary of associations between physical measures and the four variant carrier status.** Error bars indicate 95% confidence intervals of the effect sizes, and adjusted $p$-values are shown. Red lines highlight significant positive associations with FDR adjusted $p$ values < 0.05. Sex, age, and ancestry are included as covariates. Blue lines highlight significant negative associations. Abbreviations: het heterozygous, hemi hemizygous. The x-linked *GPC3* variant (X-133692376-C-T) has hemizygous carriers. ***: $p < 0.001$; **: $p < 0.01$; *: $p < 0.05$.

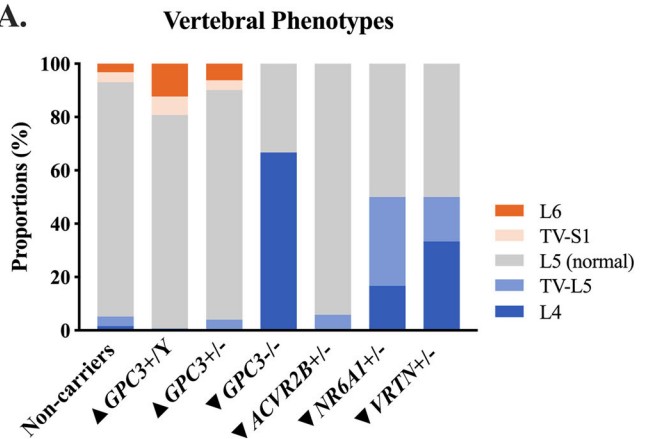

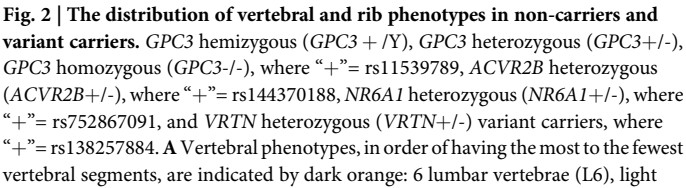

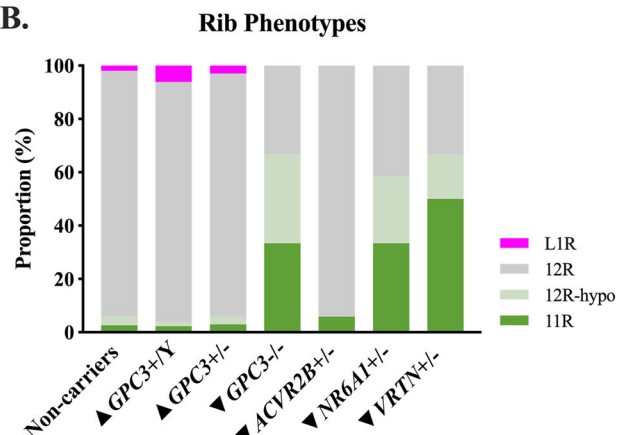

**Fig. 2 | The distribution of vertebral and rib phenotypes in non-carriers and variant carriers.** *GPC3* hemizygous (*GPC3 + /Y*), *GPC3* heterozygous (*GPC3+/-*), *GPC3* homozygous (*GPC3-/-*), where "+"= rs11539789, *ACVR2B* heterozygous (*ACVR2B+/-*), where "+"= rs144370188, *NR6A1* heterozygous (*NR6A1+/-*), where "+"= rs752867091, and *VRTN* heterozygous (*VRTN+/-*) variant carriers, where "+"= rs138257884. **A** Vertebral phenotypes, in order of having the most to the fewest vertebral segments, are indicated by dark orange: 6 lumbar vertebrae (L6), light orange: transitional S1 (TV-S1), grey: normal anatomy (L5), light blue: complete sacralisation (TV-L5), dark blue: 4 lumbar vertebrae (L4). **B** Rib phenotypes, in order of having the most to the fewest ribs, are indicated by magenta: L1 ribs (L1R), grey: 12 rib pairs (12 R), light green: hypoplastic 12th ribs (12R-hypo), and dark green: 11 rib pairs (11 R). ▲ and ▼ next to the genotype indicate whether this variant is identified through having a positive or negative association with sitting height.

for potential vertebral phenotypes. We assessed the association between the carrier status of these variants and body measures, adjusting for age, sex, and ancestry as covariates (Fig. 1).

A significant negative association was observed between sitting height and heterozygous carriers of rs752867091 in *NR6A1* (beta: −0.91, 95% confidence interval [CI]: [−1.64, −0.18], $p = 0.031$) and rs138257884 in *VRTN* (beta: −2.33, 95% CI: [−3.58, −1.08], $p = 0.0012$). In contrast, we found a positive association between sitting height and *GPC3* rs11539789 female heterozygous carriers (beta: 0.39, 95% CI: [0.23, 0.55], $p < 0.0001$), as well as hemizygous carriers (beta: 0.78, 95% CI: [0.52, 1.04], $p < 0.0001$). *GPC3* homozygous females are not included in the regression model due to the small sample size ($N = 5$). These associations were consistent across seated and standing height measurements, and the patterns align with results from the AstraZeneca PheWAS Portal (AZPheWAS)[40]. This supports the hypothesis that the selected candidate variants may be enriched for vertebral numeric variation and reflected in different height measurements. We further assessed the association between the carrier status of these variants and other body size measurements, including waist and hip circumference, but found no significant associations with these body composition phenotypes.

In terms of pulmonary function, a significant negative association was identified between forced vital capacity (FVC) and heterozygous carriers of rs144370188 in *ACVR2B* (beta: −0.38, 95% CI: [−0.64, −0.13], $p = 0.011$). Conversely, *GPC3* rs11539789 heterozygous carriers showed a positive association with FVC (beta: 0.045, 95% CI: [0.013, 0.077], $p = 0.015$), and with an even stronger positive association was observed in *GPC3* rs11539789 hemizygous carriers (beta: 0.16, 95% CI: [0.10, 0.23], $p < 0.0001$), reflecting that these variants could impact respiratory function, possibly through their effects on skeletal development. This same pattern was also observed for forced expiratory volume in 1-s (FEV1), suggesting the potential impact of these variants on overall pulmonary capacity.

Beyond these traits, we systematically examined a broad range of additional phenotypes using the AZPheWAS browser, which encompasses ~10,000 binary and ~3500 continuous traits, as well as blood metabolite measurements, clinical biomarkers, and plasma protein levels derived from UKB. This comprehensive resource enabled us to evaluate potential associations across a wide spectrum of health-related traits. No additional significant associations were identified for the four variants beyond those reported above. All observed associations are summarized in Table 1.

## Variant carrier status and skeletal phenotypes

Figure 2 shows the distributions of vertebral (Fig. 2A) and rib phenotypes (Fig. 2B) across non-carriers ($N = 500$) and variant carriers (varying sizes, $N > 5$). Table 2 summarises the odds ratio (OR) for vertebral and rib phenotypes in variant carriers compared to non-carriers.

In the 500 non-carriers, eight (1.6%) individuals have four lumbar vertebrae, 18 (3.6%) have completely sacralised L5, 439 (88%) have normal anatomy, 19 (3.8%) have transitional S1, and 16 (3.2%) have 6 lumbar vertebrae. Thirteen (2.6%) participants have 11 rib pairs, 17 (3.4%) have hypoplastic 12th ribs, 460 (92%) have 12 rib pairs, and 10 (2.0%) have L1 ribs.

In the 130 *GPC3* hemizygous males, no individual has four lumbar vertebrae, one (0.77%) individual has completely sacralised L5, 104 (80%) have normal anatomy, nine (6.9%) have transitional S1, and 16 (12%) have six lumbar vertebrae. Three (2.3%) have 11 rib pairs, two (1.5%) have hypoplastic 12th ribs, 117 (90%) have 12 rib pairs, and eight (6.2%) have L1 ribs. The odds ratio of *GPC3* hemizygous carriers having six lumbar vertebrae was significantly higher compared to non-carrier males (OR: 4.04, $p = 0.0014$). In the 274 *GPC3* heterozygous females, one (0.37%) individual has four lumbar vertebrae, 10 (3.7%) individuals have completely sacralised L5, 236 (86%) have normal anatomy, 10 (3.7%) have transitional S1, and 17 (6.2%) have six lumbar vertebrae. Eight (2.9%) have 11 rib pairs, 8 (2.9%) have hypoplastic 12th ribs, 250 (91%) have 12 rib pairs, and eight (2.9%) individuals have L1 ribs. In *GPC3* homozygous females, two (67%) individuals have four lumbar vertebrae, and one (33%) has normal anatomy. One (33%) has 11 rib pairs, one (33%) has hypoplastic 12th ribs, and one (33%) has 12 rib pairs. The odds ratio of *GPC3* homozygous carriers having four lumbar vertebrae was significantly higher compared to non-carrier females (OR: 133, $p = 9.0 \times 10^{-4}$).

In *ACVR2B* heterozygous carriers, no individuals have four lumbar vertebrae, 1 (5.9%) has completely sacralised L5, 16 (94%) have normal anatomy, no individual has transitional S1 or six lumbar vertebrae. One (5.9%) individual has 11 rib pairs, and 16 (94%) have 12 rib pairs.

In *NR6A1* heterozygous carriers, two (17%) individuals have four lumbar vertebrae, four (33%) individuals have completely sacralised L5, and six (50%) have normal anatomy; no individual has transitional S1 or six lumbar vertebrae. Four (33%) have 11 rib pairs, three (25%) have hypoplastic 12th ribs, five (42%) have 12 rib pairs, and no individual has L1 ribs. Compared to non-carriers, the odds ratio of *NR6A1* heterozygous carriers having completely sacralised L5 (OR: 14.5, $p = 6.4 \times 10^{-4}$) and 11 rib pairs

**Table 2 | Results of Fisher's exact odds ratio (OR) for vertebral and rib phenotypes in variant carriers compared to non-carriers**

| | | Genotypes | | | | | | | | | | |
| --- | --- | --- | --- | --- | --- | --- | --- | --- | --- | --- | --- | --- |
| | | ▲GPC3+/Y | | ▲GPC3+/− | | ▼GPC3−/− | | ▼ACVR2B+/− | | ▼NR6A1+/− | | ▼VRTN+/− | |
| | | OR | p value | OR | p value | OR | p value | OR | p value | OR | p-value | OR | p value |
| Vertebral phenotypes | 6 lumbar vertebrae | 4.04 | **0.0014***  | 1.31 | 0.42 | 0.00 | 1.00 | 0.00 | 1.00 | 0.00 | 1.00 | 0.00 | 1.00 |
| | transitional S1 | 1.92 | 0.21 | 0.87 | 0.86 | 0.00 | 1.00 | 0.00 | 1.00 | 0.00 | 1.00 | 0.00 | 1.00 |
| | normal anatomy | 0.58 | 0.07 | 1.08 | 0.76 | 0.08 | 0.05 | 2.76 | 0.49 | **0.16** | **3.71E-03*** | 0.17 | 0.04 |
| | complete sacralisation | 0.20 | 0.11 | 0.97 | 1.00 | 0.00 | 1.00 | 1.64 | 0.48 | **14.50** | **6.42E-04**  | 5.31 | 0.20 |
| | 4 lumbar vertebrae | 0.00 | 0.18 | 0.17 | 0.08 | 133.25 | **9.05E-04** | 0.00 | 1.00 | 14.11 | 0.01 | 35.50 | **3.42E-03*** |
| Rib phenotypes | L1 ribs | 2.44 | 0.10 | 1.09 | 0.83 | 0.00 | 1.00 | 0.00 | 1.00 | 0.00 | 1.00 | 0.00 | 1.00 |
| | 12 rib pairs | 0.80 | 0.57 | 1.16 | 0.63 | 0.06 | 0.03 | 1.72 | 1.00 | **0.07** | **3.30E-05*** | **0.05** | **1.05E-03*** |
| | hypoplasia 12th ribs | 0.68 | 1.00 | 0.81 | 0.70 | 11.26 | 0.13 | 0.00 | 1.00 | 10.36 | 0.01 | 5.84 | 0.19 |
| | aplasia 12th ribs | 0.68 | 0.76 | 0.77 | 0.70 | 13.03 | 0.11 | 1.74 | 0.46 | **15.53** | **5.09E-04** | **30.20** | **7.31E-04** |

Rows list phenotypes, while columns represent the odds ratio and p-values compared to non-carriers. OR > 1 indicates increased odds, and OR < 1 indicates decreased odds of the phenotype in carriers. GPC3+/Y are compared to male non-carriers, and GPC3−/− are compared to female non-carriers. Significant p-values, corrected for multiple comparisons, are denoted as follows: ***: $p < 0.001$; **: $p < 0.01$; *: $p < 0.05$ and shown in bold font. ▲ and ▼ next to the genotypes indicate whether this variant has a positive or negative association with sitting height.

**Fig. 3 | Associations between vertebral anomalies and the experience of pain lasting 3+ months (including headaches, neck/shoulder pain, and back pain).** Vertebral anomalies assessed include six lumbar vertebrae (6 L), transitional S1 (TV-S1), complete sacralisation (TV-L5), and four lumbar vertebrae (L4), with normal anatomy serving as the reference category. Logistic regression models were used, adjusting for sex and age as covariates. Odds ratios (ORs) are presented on a log scale, with error bars indicating 95% confidence intervals. Significant ORs >1 are highlighted in red, with * indicating significant associations (FDR adjusted one-sided p-value < 0.05) with increased pain.

(OR: 15.5, $p = 5.1 \times 10^{-4}$) are significantly higher. The odds ratios for *NR6A1* heterozygous carriers to have normal vertebral anatomy (OR: 0.16, $p = 3.7 \times 10^{-3}$) and 12 rib pairs (OR: 0.07, $p = 3.30 \times 10^{-5}$) are significantly lower than non-carriers.

In *VRTN* heterozygous carriers, two (33%) individuals have four lumbar vertebrae, one (17%) individual has completely sacralised L5, and three (50%) have normal anatomy, no individual has transitional S1 or six lumbar vertebrae. Three (50%) individuals have 11 rib pairs, 1 (17%) has hypoplastic 12th ribs, and two (33%) have 12 rib pairs, and no individual has L1 ribs. Compared to non-carriers, the odds ratios of *VRTN* heterozygous carriers having four lumbar vertebrae (OR: 35.5, $p = 3.4 \times 10^{-3}$) and 11 rib pairs (OR: 30.2, $p = 7.3 \times 10^{-4}$) are significantly higher. The odds ratios for *VRTN* heterozygous carriers to have 12 rib pairs (OR: 0.05, $p = 1.05 \times 10^{-3}$) are significantly lower than non-carriers.

Across all non-carriers and variant carriers with mixed sexes, for both vertebral and rib phenotypes, none of the comparisons between males and females showed statistically significant differences (Supplementary Table 1).

### Skeletal phenotypes and physical measures
We further explored the relationship between vertebral anatomical anomalies and physical measures (Supplementary Fig. 1). Lumbar spine curvature is included as a covariate to control for potential confounding effects for height measurements. Consistent with the AZPheWAS results[40], seated height was significantly associated with numeric variations in lumbar vertebrae. Depending on the presence of extra or reduced vertebral segments, an increase or decrease in seated height was observed in cases with six lumbar vertebrae, complete sacralisation of L5, and four lumbar vertebrae. A significant reduction in standing height is noted in individuals with complete sacralisation of L5, but not in other categories. Additionally, a reduced hip circumference is observed in those with complete sacralisation of L5.

### Experience of pain
The associations between vertebral anomalies and the experience of pain lasting 3+ months were assessed using logistic regression, adjusting for sex and age as covariates (Fig. 3). All p-values were adjusted for false discovery rate (FDR). The results indicate that having completely sacralised L5 (OR: 3.8, $p = 0.035$) and four lumbar vertebrae (OR: 7.2, $p = 0.018$) have

significantly higher odds ratios of experiencing headache than normal vertebral anatomy. Individuals with four lumbar vertebrae have a significantly increased odds ratio (OR: 4.1, $p = 0.039$) for experiencing back pain compared to those with normal anatomy. Individuals with six lumbar vertebrae have a significantly increased odds ratio (OR: 2.5, $p = 0.027$) for experiencing back pain compared to those with normal anatomy. Most rib variations are not significantly associated with long-term pain outcomes, except having 11 rib pairs has a marginal association with increased headache (OR: 3.2, $p = 0.044$) (Supplementary Fig. 2). We also investigated whether the carrier status of these candidate variants is associated with long-term pain outcomes but found no significant association (Supplementary Fig. 3). These findings suggest that vertebral configuration, rather than specific candidate genetic variants, may play a more direct role in influencing pain risk (Supplementary Fig. 4).

## Associations with health-related outcomes

Motivated by recent findings for individuals with *NR6A1* ultra-rare variants who not only displayed vertebral anomalies but also eye disorders, renal and uterine anomalies[17,18], we also assessed the impact of the genetic variants in *NR6A1*, *GPC3*, *ACVR2B*, and *VRTN* beyond vertebral-related traits. Compared to participants with a typical number of thoracic and lumbar vertebrae, participants with vertebral anomalies are more prone to disorders of the eye, digestive system, and musculoskeletal system (adjusted *p*-value < 0.05) (Supplementary Table 2). Participants identified with six lumbar vertebrae are more likely to develop H02 disorders of the eyelid (OR: 4.7, $p = 0.016$). Participants identified with transitional S1 are more likely to have disorders of H52 refraction and accommodation (OR: 6.1, $p = 0.011$). Participants with completely sacralised L5 are more likely to develop M47 spondylosis (OR: 5.2, $p = 0.0073$). Participants with four lumbar vertebrae are more likely to have K44 diaphragmatic hernia (OR: 8.7, $p = 0.0014$) and K92 digestive system disorders (OR: 7.4, $p = 0.016$).

We then investigated the association between carrier status of these candidate variants and disease traits, finding significant links for several conditions (Supplementary Table 3). Participants with the *GPC3* rs11539789 variant are more likely to develop bronchitis-related disorders (OR: 1.39, $p = 0.043$). Participants with the *NR6A1* rs752867091 variant are more likely to develop genitourinary disorders, specifically N76 inflammation of the vagina and vulva (OR: 5.00, $p = 0.019$). Participants with the *VRTN* rs138257884 variant are more likely to develop K13 disorders of the lip and oral mucosa (OR: 7.80, $p = 0.0068$), J43 emphysema (OR: 7.64, $p = 0.025$), and J45 asthma (OR: 3.04, $p = 0.029$).

Lastly, participants with the *ACVR2B* rs144370188 variant are more likely to develop M46 spondylopathies (OR: 9.93, $p = 0.0007$) and H25 senile cataract (OR: 3.55, $p = 0.013$).

## Discussion

In this study, we demonstrate the ability to characterise vertebral and rib numeric variation in the UKB using several case studies. The study leverages genetic and imaging data from the UKB to investigate the genetic factors influencing vertebral and rib numeric variation, focusing on specific variants with potential effects on vertebral development. Our findings included significant associations between novel genetic variants in candidate genes, imaging-derived skeletal phenotypes, and body size measurements, suggesting that these variants may play a role in shaping the development of vertebral column. We were able to implicate three genetic variants in three candidate genes (*GPC3*, *VRTN*, *NR6A1*) with human vertebral phenotypes. Additionally, we identified links between vertebral anomalies and chronic pain, offering insights into the potential clinical impact of these variations.

Chronic back pain is a major cost to health systems worldwide. In Australia, chronic back pain is estimated to cost the health system $4.8 billion each year[41]. Costly back surgery can be avoided in some cases, but there are no molecular biomarkers and related specific back anomalies to indicate patients that require intensive imaging analysis. The variants we have identified have MAF ranging from 0.003 to 0.46% and although they

represent a small fraction of potential causes for back pain, they lead to the first potential genetic markers for back pain.

Previous studies have reported substantial variability in numeric variants and transitional lumbosacral vertebrae, ranging from 3.3 to 35.6%[3–5,7,42–47], with differences attributed to factors such as sample selection[5,7,48], imaging modalities[25,49], vertebral enumeration methods[50], and classification criteria[44,51]. Numeric variations, such as having four or six lumbar vertebrae, are much rarer phenotypes that are not frequently reported. Although 3D CT is considered the gold standard for assessing skeletal morphology large datasets, coupled with genetic data, are not readily available. The UKB remains the largest dataset with genetic and body imaging data available. We demonstrate that by leveraging (combining) the DXA and Dixon MRI, it is feasible to detect vertebral variation. This is despite the high-resolution DXA data only being available for the lumbar region. This may lead to an underestimation of short ribs. This limitation also makes it challenging to accurately assess transitional thoracic-lumbar vertebrae. For example, cases labelled as having 'four lumbar vertebrae' in the current study may, in some instances, correspond to individuals with 11 thoracic and five lumbar vertebrae. We addressed this limitation by fusing multimodal Dixon MRI body images for more accurate assessment. Additionally, many previous studies exhibit sampling bias, often focusing on individuals with back pain or disorders[43,52], who are more likely to present with atypical anatomy. Due to incomplete health records, identifying a reliable sample of healthy controls within the UKB is challenging. To establish a representative baseline, we selected non-carriers of the four identified putative vertebral diversity genetic variants to reflect the broader population characteristics.

Compared to non-carriers, we found individuals carrying variants in *GPC3*, *NR6A1*, and *VRTN* exhibited an enrichment of vertebral and rib anomalies, including numeric variations and transitional vertebrae, with incomplete penetrance and phenotypic heterogeneity. *GPC3* is known to be associated with the presence of six lumbar vertebrae[53], but the variant rs11539789 identified in this study has not been previously associated with this phenotype, highlighting its potential new role in vertebral development, which requires further investigation. *NR6A1* has recently been found to be associated with numerical variations in the lumbar vertebrae and ribs in ultra-rare clinical studies[17,18]. In this study, we identified the novel variant rs752867091 in *NR6A1* associated with vertebral and rib anomalies. The *VRTN* gene, previously identified as being associated with vertebral number in animal models[34,35], has now, for the first time, been observed to be associated with abnormal skeletal development in humans in this study. This discovery marks a significant step forward in translating insights from animal models to human biology. The genetic regulation of vertebral column development has long been studied in animal models, but its direct application to human development remained unclear due to the inherent complexities. By leveraging the genetic and imaging data from the UKB, we were able to link genomic variations to specific changes in vertebral column development in humans.

During our initial variant search, we used body size measurements, particularly sitting height, to identify candidate variants. We hypothesized that variants in *NR6A1*, *VRTN*, and *ACVR2B*, which are associated with reduced sitting height, may indicate fewer vertebral segments, while *GPC3* variants, associated with increased sitting height, may correspond to additional vertebral segments. We chose sitting height as the most relevant phenotype because it directly reflects the length and development of the vertebral column, minimizing the confounding influence of limb length observed in overall stature. This finding was validated through the associations we were able to identify between vertebral phenotypes and body size measurements, suggesting that body size measurements can serve as useful proxies for identifying vertebral anomalies when direct anatomical data are unavailable. However, these proxies may not fully capture the complexity of vertebral anatomy, and future studies incorporating more direct imaging-based phenotyping are needed to provide greater specificity. Our work has shown that this is feasible in the UKB.

**Table 3 | Baseline characteristics of variant carriers and noncarriers in the UK Biobank of the whole exome sequencing (WES) data final release and the imaging sub cohort**

| | All participants | GPC3 carriers | NR6A1 carriers | VRTN carriers | ACVR2B carriers | Control baseline |
|---|---|---|---|---|---|---|
| | | X-133692376-C-T rs11539789 c.1285G > A p.Val429Met | 9-124524718-T-C rs752867091 c.1351+3A > G | 14-74357184-G-A rs138257884 c.401G > A p.Arg134His | 3-38483237-C-T rs144370188 c.1444C > T p.Arg482Trp | Absence of all four of the variants |
| **UKB WES data final release** | | | | | | |
| Sample size, N | 406,069 | 3046 Hom/Het/Hemi 6/2098/942 | 114 (Het) | 37 (Het) | 138 (Het) | 402,735 |
| Female (%) | 54.10% | 69.10% | 57.90% | 51.40% | 47.10% | 54.00% |
| Age (years), mean (SD) | 56.5 (8.1) | 56.5 (7.9) | 55.6 (7.9) | 56.9 (8.0) | 57.5 (7.7) | 56.5 (8.1) |
| **Imaging visit (2014+)** | | | | | | |
| Sample size, N | 76,277 | 407 Hom/Het/Hemi 3/274/130 | 12 (Het) | 6 (Het) | 17 (Het) | 500 |
| Female (%) | 51.60% | 68.10% | 58.30% | 50% | 41.20% | 50.00% |
| Age (years), mean (SD) | 65.5 (7.8) | 65.1 (7.6) | 59.8 (6.6) | 68 (7.4) | 65.8 (6.9) | 65.6 (7.8) |

Our study has several shortcomings. There may be other confounding factors, as individuals with altered sitting height do not necessarily exhibit numeric variations in lumbar vertebrae. To account for this, we included the curvature of the lower spine as a covariate when testing for associations. Other relevant phenotypes might include vertebral size, intervertebral disc space, or other musculoskeletal congenital pathologies, such as block vertebrae. For the *ACVR2B* candidate variant, we did not observe numeric variations in vertebral or rib anomalies compared to non-carriers, despite its known association with increased thoracic vertebrae and vertebral transformations in animal studies[54]. It is possible that homeotic changes are present without a change in the overall vertebral number, which may be challenging to accurately assess given the limitations of the available imaging data.

We further explored the relationship between vertebral anomalies and chronic pain, including headaches, neck/shoulder, back, and hip pain. The presence of four lumbar vertebrae or completely sacralised L5 was associated with increased incidence of experiencing headache, while individuals with six lumbar vertebrae were more likely to report chronic back pain. These results align with previous reports suggesting that anatomical variations in the spine are associated with lower back pain[55]. Importantly, these associations were observed independently of the specific genetic variants studied, suggesting that vertebral configuration, rather than the specific genetic variants, may play a more direct role in determining pain risk.

Our study reveals that vertebral anomalies frequently co-occur with abnormalities in multiple body systems. Some of these associations were previously implicated in other congenital conditions involving vertebral abnormalities. For instance, we discovered that having four lumbar vertebrae is associated with increased likelihood of developing congenital diaphragmatic hernia, a rare condition also observed in congenital syndromes involving skeletal malformations[21,39,56]. Our study uncovered associations between this *GPC3* variant and pulmonary function measures, including FVC and FEV1, suggesting potential impacts on respiratory function. *GPC3* was known to play a crucial regulator of the Hedgehog pathway, which is essential for pulmonary development[57]. Furthermore, similar to our findings, previous research has linked *NR6A1* to abnormalities in the female genital tract and uterine malformations[17] and eye disorders[18]. These multifaceted findings highlight the interconnected nature of developmental processes in the human body but also point to specific patterns for each gene, which give clues to their neurodevelopmental program function. Future research incorporating more accurate and detailed diagnostic information could further explore these diseases in a more comprehensive way.

Despite emerging evidence supporting the involvement of *GPC3* and *NR6A1* in skeletal development from both human and experimental studies[13,17,18,20,21], one of the limitations of the current study is that we were unable to replicate the specific rare variants identified here in another independent cohorts. This is largely due to their rarity and the lack of vertebral-specific phenotypes or imaging data in other large-scale resources.

We chose to focus on specific candidate genes and protein-coding variants genetic contributors to vertebral development to establish whether this was feasible in the UKB and whether such variation even exists, as they may experience substantial selection pressure leading to their suppression or even elimination in the population. A limitation of our study is that deriving vertebral and rib phenotypes was extremely labour-intensive, especially because we used non-ideal imaging data, which required modal fusion to allow analysis. This effort constrained the scalability of phenotyping, and although we analysed over 900 participants, statistical power remained limited for some downstream analyses, particularly secondary outcomes such as pain. Future research will integrate fully automatic deep learning-based imaging methods with genome-wide association studies. This approach will enable a more comprehensive investigation into the genetic architecture of vertebral development, uncovering broader genetic factors that influence vertebral variation at the population level and on an evolutionary scale. Our work has demonstrated that this is feasible, additionally uncovering novel biology and potential biomarkers for pain.

## Methods
### UK Biobank cohort
The UKB is a large population-based database consisting of over 500,000 participants, primarily United Kingdom residents aged 40–69 years at recruitment[58]. Analysis in this study leveraged genetic, imaging, clinical, and lifestyle data from the UKB cohort. In this study, we used whole-exome sequencing (WES) data from the final exome release (Data-Field: 23157) of over 469,000 participants (See Table 3 for cohort description). We excluded the following individuals: (1) related individuals based on genetic relatedness, retaining one individual from each related pair, prioritizing those carrying a variant of interest; (2) participants who withdrew from the UKB; and (3) participants with a sex mismatch (reported sex did not match genetic recorded sex). After applying these quality control exclusions, the final dataset included 406,069 UKB participants. Access to all UKB data was granted on June 18th, 2019, application #36610. This study was approved by the Walter and Eliza Hall Institute of Medical Research (WEHI), Human Research Ethics Committee (HREC reference 17/09LR).

## Gene and variant selection

To identify genetic variants that may have potential effects on vertebral numeric variation, we applied the following steps to select candidate genes and variants (refer to Supplementary Fig. 5 for the full list of candidate genes/variants and the selection process).

A gene panel was designed utilizing known clinical or phenotypic associations. Genes related to numeric variations of the thoracic and lumbar vertebrae were collected from the HPO and the MGI phenotype databases, along with additional genes identified in our previous mouse studies[13]. Associated genes were extracted from phenotype categories, including supernumerary vertebrae (HP:0002946), increased lumbar vertebrae number (MP:0004650), decreased lumbar vertebrae number (MP:0004647), increased thoracic vertebrae number (MP:0004651), and decreased thoracic vertebrae number (MP:0004648). Additionally, the *NR6A1* gene, identified in prior research as controlling vertebral number and segmentation in the trunk region of mice[13], was also included. A total of 51 genes were identified (as of HPO and MGI updates in April 2024).

In the second step, we sought to identify specific candidate variants within the gene panel that may contribute to vertebral numeric variation. To facilitate this, we accessed variant-phenotype association summary statistics from the AZPheWAS (https://azphewas.com, 470k (v5) public version)[40]. The analysis by AZPheWAS is based on a subset of approximately 420,000 high-quality exome sequences, predominantly from unrelated study participants of European ancestry, released by the UKB. These were used to evaluate the association between protein-coding variants and phenotypes through variant-level and gene-level phenome-wide association studies (PheWAS). Given that UKB does not provide information on anatomical variations of the vertebral column directly, we explored phenotypes that might be indirectly related. Body size measurements, including human height (e.g., standing and sitting height), are highly heritable and polygenic traits that have been extensively studied in terms of their genetic basis[59]. Specifically, sitting height (Data-Field: 20015), which measures the length of the torso, is more likely to be a proxy for the numerical variation of the vertebral column. We selected sitting height as the most relevant phenotype because it directly reflects the length and development of the vertebral column, minimising the confounding influence of limb length seen in overall stature.

To identify specific candidate variants contributing to numerical variations in human vertebrae, we used AZPheWAS data and applied the following criteria: (1) restrict the analysis to 51 genes of interest (2334 variants); (2) retain less common and rare alleles with a MAF ≤ 0.01 within the UKB cohort (2194 variants); (3) remove synonymous variants (1499 variants); (4) identify variants associated with body size–related measurements within ICD-10 Chapter XXI (Factors influencing health status and contact with health services) at $P < 1 \times 10^{-5}$ to capture a broader range of potential candidates (13 variants; Supplementary Table 4); and (5) further narrow to variants where sitting height showed the strongest association (4 variants), excluding those where associations were primarily driven by standing height, weight, body fat percentage, or trunk fat mass.

We focused on rare variants (MAF ≤ 1%) because these are more likely to lead to congenital anomalies in vertebral development, such as numeric changes in vertebral or rib number. This also aligns with the reported prevalence of congenital numeric variations, such as lumbar and cervical ribs, which each occur in <1% of the population. Sitting height was chosen as a key proxy phenotype because our aim was to identify variants associated with numeric anatomical anomalies, such as variation in vertebral number. In contrast to other body measurements, sitting height is more directly influenced by the spine and therefore better suited to capture congenital differences in vertebral segmentation rather than proportional differences in overall skeletal size (e.g., taller stature or longer limbs).

We further examined the candidate genes and variants by gathering additional information on population allele frequency, clinical significance, phyloP conservation scores, and other relevant data from key public databases, including the OMIM[36], gnomAD (version 4.1)[60], ClinVar[37], and the GWAS Catalog[61]. These resources provide extensive clinical and population-level insights, supporting a more comprehensive analysis of the selected genes and genetic variants.

## Sample selection

Carriers of the four candidate variants were identified using the UKB WES data, specifically the population-level exome OQFE variant dataset from the 450k release (Field ID: 23157; genome build GRCh38), provided by UKB[62]. These data were accessed via the "Genomics" section of the UKB Research Analysis Platform cohort browser (https://ukbiobank.dnanexus.com/). We retrieved a subgroup of 442 candidate variant carriers with body imaging data (detailed in the next section) available for vertebral and rib phenotyping (Table 3). None of the participants in the study carried more than one candidate variant.

Due to the large manual curation burden, we selected 500 non-carriers to serve as a control group representing the characteristics of the general population, allowing us to assess whether vertebral anatomical anomalies are enriched among variant carriers. Non-carriers (250 female and 250 male) were selected based on the following criteria: (1) absence of all four of the variants, (2) unrelated individuals based on genetic relatedness, (3) exclusion of participants who withdrew from UKB, (4) restriction to individuals of European genetic ancestry based on Pan-UKBB (https://pan.ukbb.broadinstitute.org)[63], (5) age-matched to the variant carriers (using the age at the time of imaging session), and (6) availability of multimodal body imaging data.

## Image-based vertebral and Rib phenotyping

**UK Biobank body imaging data.** Dual-energy X-ray Absorptiometry (DXA) images (Data-Field: 20158) and whole-body Dixon MRI images (Data-Field: 20201) from the UKB were used to assess the anatomical variations of the vertebral column and associated skeletal structures. All participants were imaged according to a harmonised UKB DXA and body MRI protocol[64]. In brief, DXA quantifies bone mineral density and body composition by measuring the absorption of X-rays at two different energy levels[65]. DXA images of the whole body, lumbar spine (L1-L4), and lateral spine (T4-L4) were acquired on a GE-Lunar iDXA scanner. Dixon MRI is a magnetic resonance imaging technique that separates water and fat signals, allowing for soft tissue characterisation[66]. Dixon MRI scans were acquired using a Siemens Aera 1.5-T scanner with the dual-echo Dixon Vibe protocol, covering the neck to knees (1.1 m total) in six volumes. The resolution varied slightly across volumes, ranging from 2.23 × 2.23 × 3-4.5 mm³. For detailed imaging parameters, refer to the UKB imaging protocol and information reported in previous studies[64,67,68].

**Image pre-processing and alignment.** Body images were pre-processed and aligned using a deep-learning-based multi-modal image-matching contrastive framework[69] with modifications. A total of 20,202 pairs of whole-body DXA and MRI scans from the UKB were randomly selected and pre-processed according to github.com/rwindsor1/UKBiobankDXAMRIPreprocessing. Fat-only and water-only Dixon MRI scans, acquired in six volumes, were stitched into two 3D volumes. The centre of mass of the 3D volumes was located according to the coronal slice intensity histogram in the fat-only scans. Five neighbouring 2D slices on each side of the centre of mass were extracted. All scans are resampled to be isotropic and cropped to 800 × 300 pixels for DXA and 501 × 224 pixels for MRI scans with 2.2 mm isotropic pixel spacing.

The pre-processed images were split into 80/10/10% for model training, testing, and validation, respectively. Self-supervised contrastive learning was implemented, where two convolutional neural networks take a pair of 2D DXA-MRI images as input. The spatial feature maps of each modality were then convolved and correlated to find the best alignment. The maximum correlation of the dense spatial feature maps was used to perform the registration translation for aligning the two images. Detailed information on network architecture and implementation can be found on github.com/rwindsor1/biobank-self-supervised-alignment.

**Vertebral and rib anatomical variation annotation**. After DXA and Dixon MRI scans were aligned, manual annotation was performed to assess the vertebral and rib numbers. C2 or T1 landmarks are identified for accurate vertebral numbering. A small number of Dixon MRI scans have varied field-of-views, where, in general, cervical vertebrae could be observed from C2 onwards, but not always. In cases where C2 can be clearly identified, the number of vertebrae was counted down from C2 on the whole body MRI scans, assuming seven cervical and 12 thoracic vertebrae (Supplementary Fig. 6A)[3]. This standard enumeration method was chosen because thoracolumbar transitional vertebrae cannot be reliably identified using the UKB imaging data (with the high-resolution image only covering the lumbar region).

In the absence of C2 on the MRI scans, the intervertebral discs were manually labelled on the 3D MRI scans first, and the labelled masks were overlayed on the pre-registered whole-body DXA scans. The first thoracic vertebrae (T1) were identified on the DXA image by locating the first rib pairs. The number of presacral segments was counted starting from T1 down to the last presacral segment (Supplementary Fig. 6B).

The vertebral phenotypes derived in the current study consist of the five categories (Supplementary Fig. 7A): (1) six lumbar vertebrae (characterised by an additional lumbar segment that displays features such as squaring of the vertebral body); (2) lumbarised S1 (a sacral segment exhibits lumbar-like characteristics, partially completely separated from the sacrum); (3) normal (the standard configuration of five lumbar vertebrae; (4) complete sacralisation (complete fusion of the fifth lumbar vertebra with the sacrum); and (5) four lumbar vertebrae (a reduced count of lumbar vertebrae). The rib phenotypes consist of the following four categories (Supplementary Fig. 7B): (1) extra ribs on L1; (2) 12 rib pairs; (3) hypoplasia of the 12th ribs (underdeveloped or reduced size of the 12th ribs); (4) aplasia of the 12th ribs (the absence of the 12th ribs).

In addition to numeric variations, the curvature of the lumbar spine was measured using the posterior tangent method[70], measured as the angle formed between the posterior tangent lines L1 and L5 vertebral bodies on the lateral spine DXA.

**Physical measures and health-related outcomes in UKB**. Relevant physical measures from the UKB were retrieved to examine genetic and phenotypic relationships. These included body size measurements such as standing height (Data-Field: 50), seated height[51], sitting height (Data-Field: 20015), waist circumference (Data-Field: 48), and hip circumference (Data-Field: 49), as well as pulmonary function indicators, specifically forced expiratory volume in 1-s (FEV1, Data-Field: 3063) and forced vital capacity (FVC, Data-Field: 3062).

Pain experience was assessed using responses from the question posed at the assessment centre. For participants who answered "yes," additional information was collected on those who reported specific pain types lasting for more than three months[71]. We focused our analysis on a subset of chronic pain types, including headaches (Data-Field: 3799), neck/shoulder pain (Data-Field: 3404), back pain (Data-Field: 3571), and hip pain (Data-Field: 3414).

For exploratory analysis, we retrieved health related outcomes (Category: 1712) in the format of ICD-10-coded diseases derived from linkage to primary care data (Category: 3000), hospital inpatient data (Category: 2000), death register records (Data-Field: 40001, 40002), and self-reported medical conditions at baseline or follow-up (Data-Field: 20002). We focused our analysis on 600 ICD-10 diagnostic terms within nine ICD disease chapters: musculoskeletal system and connective tissue disorders, congenital disruptions and chromosomal abnormalities, digestive system disorders, eye and adnexa disorders, ear and mastoid process disorders, genitourinary system disorders, nervous system disorders, respiratory system disorders, and circulatory system disorders.

**Statistics and reproducibility**. We firstly evaluated whether variant carrier status was associated with phenotypes of interest. Associations with continuous phenotypes, such as body size measurements and pulmonary measures, were assessed using general linear models, adjusting for age at recruitment (Data-Field: 21022), sex (Data-Field: 31), and genetic ancestry using the first 10 principal components (PCs). Associations with binary pain phenotypes were assessed using logistic regression models, also adjusting for age, sex, and ancestry (PC1-10). Odds ratios (ORs) or effect sizes, along with 95% confidence intervals (CIs), were calculated. All p-values are adjusted for the FDR, with significance defined as adjusted $p$-values $< 0.05$.

Statistical analyses were conducted to evaluate associations between vertebral and rib phenotypes and various UKB measures, including reported pain and physical measurements. Logistic regression models were used to assess the relationships between vertebral and rib phenotypes and body size measurements, as well as reported pain. Age at imaging and sex were included as covariates to control for potential confounders. Lumbar spine curvature, as measured by the curvature between the first and last lumbar segment using the posterior tangent method, was included as an additional covariate when testing for associations with height measurements[70]. Logistic regression models were used to compare the associative effects of genotypes and vertebral phenotypes with experience of pain. Binary variables were created for both predictors, and the results were visualized side by side as odds ratios with 95% confidence intervals to compare their relative effects. All $p$-values are adjusted for the FDR, with significance defined as adjusted $p$-values $< 0.05$.

We explored associations between health-related outcomes and vertebral phenotypes using logistic regression, adjusting for sex. We also examined associations between these health outcomes and genetic variants. $P$-values were adjusted using FDR correction for vertebral phenotypes and genetic variants within each ICD-10 subcategory, with a significance threshold of $p < 0.05$ after FDR correction.

## Reporting summary
Further information on research design is available in the Nature Portfolio Reporting Summary linked to this article.

## Data availability
All WES, physical measurements, reported pain data, and imaging data described in this paper are publicly available to registered researchers through the UKB data-access protocol. Additional information about registration for access to the data are available at: https://www.ukbiobank.ac.uk/enable-your-research/apply-for-access. Further information about the WES data is available at https://www.ukbiobank.ac.uk/media/najcnoaz/access_064-uk-biobank-exome-release-faq_v11-1_final-002.pdf. Detailed information about UK biobank imaging is available at https://biobank.ndph.ox.ac.uk/ukb/ukb/docs/body_mri_explan.pdf for body MRI and https://biobank.ctsu.ox.ac.uk/crystal/crystal/docs/DXA_explan_doc.pdf for DXA imaging. The image pre-processing and alignment packages are publicly available at: https://github.com/rwindsor1/UKBiobankDXAMRIPreprocessing. https://github.com/rwindsor1/biobank-self-supervised-alignment. Numerical source data underlying all graphs in the manuscript can be found in supplementary data file.

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

## Acknowledgements

This work was supported by an Australian National Health and Medical Research Council (NHMRC) Investigator Grant to M.B. (GNT1195236). We sincerely thank the AstraZeneca PheWAS project teams for providing access to summary statistics and other public data resources. This work was also made possible through the Victorian State Government Operational Infrastructure Support and Australian Government National Health and Medical Research Council Independent Research Institute Infrastructure Support Scheme. This research has been conducted using the UK Biobank Resource (https://www.ukbiobank.ac.uk/enable-your-research/manage-your-project) under Application Number 36610. We would like to thank the participants and their families, without whom these studies would not have been possible. We also like to thank the technical support provided by WEHI's research computing team.

## Author contributions

Conceptualization: M.B., Z.S., and J.H. Investigation: M.B., Z.S., and J.H. Data curation: Z.S., J.H., and L.G.F. Formal analysis: Z.S. and J.H. Methodology: M.B., Z.S., and J.H. Validation: Z.S., J.H., E.M., and M.B. Resources: E.M. and M.B. Writing—original draft: Z.S. and J.H. Writing—review and editing: Z.S., J.H., L.G.F., E.M., and M.B. Funding acquisition: M.B. Supervision: M.B.

## Competing interests

The authors declare no competing interests. M.B. is an Editorial Board Member for Communications Biology, but was not involved in the editorial review of, nor the decision to publish this article.
