## [Transparent Peer Review file · Communications Biology]

Identification of novel vertebral development factors through UK Biobank candidate gene search and body imaging analysis

Corresponding Author: Professor Melanie Bahlo

This manuscript has been previously submitted at another journal. This document only contains information relating to versions considered at Communications Biology.

Version 0:

Reviewer comments:

Reviewer #2

(Remarks to the Author)

Sun et al identified novel rare variants in genes GPC3, NR6A1, and VRTN associated with developmental abnormalities in the spine using genetic and imaging data from the UK Biobank.

The authors carried out an incredibly tedious work to manually annotate the vertebral and rib numbers. Besides, they took advantage of the highly valuable source of the UK Biobank to mine on genetic associations to this under investigated trait. However, several concerns arise after the review of the manuscript:

The authors independently evaluated four variants located in four genes to be associated with changes in vertebral number. Did they identify any individual who carried more than one of these variants? This needs to be clarified in the results.

Once the authors selected the three genes associated with the vertebral deformities, did they re-analysed the common variants linked to these genes for any correlation with either the number of vertebrae or the proxy trait of sitting height? The authors report in the introduction that up to 36% of the population present vertebral abnormalities, a frequency much higher than the very rare variants selected in this study.

The authors analysed other health-related outcomes associated with the selected variants. Taking into account that most of SNPs/genes identified here were linked to metabolic disorders in other GWAs (i.e. type 2 diabetes and lipid metabolism), it would be of interest to include these metabolic traits in the health-related analysis performed for these SNPs.

Some of the analysed SNPs were predicted to be deleterious using software like SIFT, did the authors investigated whether carriers of these variants presented a clinical phenotype compatible with the rare diseases known to be caused by mutations in the investigated genes?

Although the work presented here is very valuable, replication in a separate cohort would be needed to confirm the association of the proposed variants with vertebral development.

Reviewer #3

(Remarks to the Author)

The authors have used whole exome sequence (WES) from 469,000 UK Biobank participants to perform a candidate gene (n=51 genes) study identifying variants contributing to vertebral abnormalities. This work is highly pertinent to many groups working with UK Biobank developing machine learning methods for automated reading of images, and this study uses the Oxford group code.

The authors used proxy measures such as sitting to standing height measure to assess vertebral number abnormalities, and sought rare variants (MAF < 0.01) associated with sitting height having variants present in at least 5 individuals before examining further databases and clinical information. Some of this information has been manually curated.

The work some years in the making and the authors have used knowledge of mouse genetics to shed light on genetic influence for vertebral abnormalities in humans, maintaining power by selecting candidates a priori.

Major comments:-

1. It would be helpful to have a title more reflective of the contents. There are many chronic pain GWAS reported from UK Biobank including those on back pain. I suggest the title be re-configured to contain the methodological details 'candidate gene' and perhaps 'whole exome sequencing' around vertebral anomalies, and back pain be removed.
2. Methods - there is a lack of detail around identifying the variants associated with (multiple) phenotypes. What did the manual curation amount to? What methods were used to test association with variants? Please provide details
3. What software was used to call the variants? Were CNVs considered as well as SNVs?
4. Multiple testing - how has this been taken into account? Particularly in the pain section, once multiple phenotypes have been considered, the level usual level of significance will likely not be met. Clarification would be appreciated.

Minor comments:-

1. line 75 - please clarify how this WES study of candidate genes is proof of principle for GWAS - or remove the statement
2. from 51 candidate genes 4 were selected containing variants. How was this done - simply by association with sitting height? Methods seem very sparse at this point, please expand.

Typos

Line 91/2 should read 'numerical variation'

Experiences of pain should read 'experience of pain'

Version 1:

Reviewer comments:

Reviewer #2

(Remarks to the Author)

The authors have addressed properly all the concerns arisen during the review process and made appropriate changes in the original manuscript.

In my opinion, the manuscript now is ready for publication.

Reviewer #3

(Remarks to the Author)

The authors have addressed the comments.

We thank both reviewers for their thoughtful and constructive comments. The revisions are outlined below in the point-by-point responses.

Reviewer Comments to Authors:

Reviewer #2 (Remarks to the Author):

Sun et al identified novel rare variants in genes GPC3, NR6A1, and VRTN associated with developmental abnormalities in the spine using genetic and imaging data from the UK Biobank. The authors carried out an incredibly tedious work to manually annotate the vertebral and rib numbers. Besides, they took advantage of the highly valuable source of the UK Biobank to mine on genetic associations to this under investigated trait.

However, several concerns arise after the review of the manuscript:

Comment #1: The authors independently evaluated four variants located in four genes to be associated with changes in vertebral number. Did they identify any individual who carried more than one of these variants? This needs to be clarified in the results.

Response: We thank the reviewer for this helpful comment. No participant in our study carried more than one of the candidate variants. We have added this clarification to the Methods section (lines 520-521) under “Sample Selection” as the following:

‘None of the participants in the study carried more than one candidate variant.’

Comment #2: Once the authors selected the three genes associated with the vertebral deformities, did they re-analysed the common variants linked to these genes for any correlation with either the number of vertebrae or the proxy trait of sitting height? The authors report in the introduction that up to 36% of the population present vertebral abnormalities, a frequency much higher than the very rare variants selected in this study.

Response: We thank the reviewer for this thoughtful comment. We would like to clarify that 36% is the upper limit reported for transitional vertebrae, while the prevalence of vertebral/rib number variation is much lower. The exact estimates vary between studies depending on methodology (e.g., postmortem or different imaging modalities). For instance, lumbar ribs are extremely rare, with an estimated prevalence of ~1%. In this study, we focused on the more severe forms of vertebral abnormalities, specifically numeric differences (extra or fewer lumbar segments). The rationale is that the UK Biobank is an aging population, in which many mild or transitional

vertebral changes may arise from environmental or degenerative factors rather than congenital risk factors. To better capture the genetic contribution to skeletal anatomical variation, we therefore concentrated on rare variants ($MAF \leq 1\%$) which are more likely to be penetrant and linked to more overt congenital phenotypes.

To complement this rare variant analysis, we have now added Supplementary Table 4, which lists both rare and common variants in our panel genes that show significant associations with body size measurements in the AZ PheWAS dataset. Among this expanded table, only four variants met our frequency threshold ($MAF \leq 1\%$), making them suitable for association analysis with congenital numeric variation.

We added this reasoning into the Introduction (lines 38-40; lines 92-94) and Methods (lines 497-505) as the follows:

‘...numerous clinical and anthropologic studies have reported the presence of anatomical variation such as transitional vertebrae in 3–36% of the population, along with rib variations (3–7). In addition, true numeric changes are rarer, with lumbar ribs observed in roughly 1% of individuals (8).’

‘We focused on the more severe forms of vertebral abnormalities as they are more likely to present as congenital anomalies with visible changes on current imaging modalities.’

‘We focused on rare variants ($MAF \leq 1\%$) because these are more likely to lead to congenital anomalies in vertebral development, such as numeric changes in vertebral or rib number. This also aligns with the reported prevalence of congenital numeric variations, such as lumbar and cervical ribs, which each occur in $<1\%$ of the population.’

Comment #3: The authors analysed other health-related outcomes associated with the selected variants. Taking into account that most of SNPs/genes identified here were linked to metabolic disorders in other GWAs (i.e. type 2 diabetes and lipid metabolism), it would be of interest to include these metabolic traits in the health-related analysis performed for these SNPs.

Response: We thank the reviewer for this valuable suggestion. The AstraZeneca PheWAS Portal has comprehensively assessed potential associations at both the SNP and gene levels. In the manuscript, we report associations for the four variants of interest across the full range of

phenotypes available in the portal, which includes ~10,000 binary and ~3,500 continuous traits, as well as blood metabolite measurements, clinical biomarkers, and plasma protein levels derived from the UK Biobank baseline data. All observed associations, including those with height and pulmonary function, are presented in Table 1 of the manuscript, and no other significant associations were identified.

We added this notes to the Results section (lines 169-174) as the following:

“Beyond these traits, we systematically examined a broad range of additional phenotypes using the AZPheWAS browser, which encompasses ~10,000 binary and ~3,500 continuous traits, as well as blood metabolite measurements, clinical biomarkers, and plasma protein levels derived from UKB. This comprehensive resource enabled us to evaluate potential associations across a wide spectrum of health-related traits. No additional significant associations were identified for the four variants beyond those reported above. All observed associations for the four variants are summarized in Table 1.”

Comment #4: Some of the analysed SNPs were predicted to be deleterious using software like SIFT, did the authors investigated whether carriers of these variants presented a clinical phenotype compatible with the rare diseases known to be caused by mutations in the investigated genes?

Response: We thank the reviewer for this comment. As noted in the original submission, we had already investigated this question through two complementary analyses presented in the manuscript: (i) associations between vertebral phenotypes and health-related outcomes (Table S2), and (ii) associations between health-related outcomes and the four variant carrier statuses (Table S3). These analyses showed modest associations with common traits (e.g., musculoskeletal, respiratory, and eye disorders), but no enrichment for rare or severe clinical phenotypes.

In addition, we examined clinical records for variant carriers, which were consistent with these findings and did not reveal severe or syndromic manifestations. Overall, we did not observe strong evidence of severe disease phenotypes among these variant carriers.

Comment #5: Although the work presented here is very valuable, replication in a separate cohort would be needed to confirm the association of the proposed variants with vertebral development.

Response: We agree that replication in an independent cohort would be highly valuable. To explore this further, we examined these variants in several large-scale resources, including the GWAS Catalog (<https://www.ebi.ac.uk/gwas/>), FinnGen PheWeb (<https://r6.finngen.fi/>), and BioBank Japan PheWeb (<https://pheweb.jp/>). At the SNP level, however, these specific variants were either not represented in the databases due to their rarity or do not show significant associations with body measurements. Note that most available biobank resources do not provide phenotypes with the same level of depth as the UK Biobank, particularly lacking sitting height or vertebral-specific measures and imaging data, which limits the ability to perform direct replication analyses for these traits.

At the gene level, however, there is converging support for the role of these genes in growth and vertebral biology. Other variants in *NR6A1*, *VRTN*, and *GPC3* (distinct from the rare variants selected in our study) are shown to have significant associations with standing height, reported in the GWAS Catalog, FinnGen, or BioBank Japan PheWeb. In addition, two recent studies have described rare pathogenic variants in *NR6A1*: one reporting a novel autosomal dominant oculo-vertebral-renal (OVR) syndrome characterized by colobomatous microphthalmia, missing vertebrae, and congenital kidney abnormalities (PMID: 37815251), and another identifying *NR6A1* variants as a cause of congenital renal, vertebral, and uterine anomalies (doi: 10.1101/2025.01.08.24319478). We note that the authors cite our preprint (this manuscript: <https://doi.org/10.1101/2025.02.13.25322190>) ‘Using a combination of imaging and genetic data, Sun et al. recently reported that *NR6A1* was a key gene associated with differences in vertebral number⁴³.’ as part of their evidence and we also cite their papers in our manuscript (references 18).

Together, these independent genetic findings reinforce the biological plausibility of our results. However, we hope that our findings provide evidence using population data supporting the involvement of these genes in vertebral patterning, thereby complementing the emerging clinical and experimental data.

We have expanded our Discussion (lines 397-401) to further address this limitation of lacking replication at the variant level due to the rarity of these variants as the following:

‘Despite emerging evidence supporting the involvement of *GPC3* and *NR6A1* in skeletal development from both human and experimental studies (13, 17, 18, 20, 21), one of the limitations of the current study is that we were unable to replicate the specific rare variants identified here in another independent cohorts. This is largely due to the rarity of the variants as well as the lack of vertebral-specific phenotypes or imaging data in other large-scale resources that are required to identify the phenotypic effects.’

Reviewer #3 (Remarks to the Author):

The authors have used whole exome sequence (WES) from 469,000 UK Biobank participants to perform a candidate gene (n=51 genes) study identifying variants contributing to vertebral abnormalities. This work is highly pertinent to many groups working with UK Biobank developing machine learning methods for automated reading of images, and this study uses the Oxford group code.

The authors used proxy measures such as sitting to standing height measure to assess vertebral number abnormalities, and sought rare variants (MAF < 0.01) associated with sitting height having variants present in at least 5 individuals before examining further databases and clinical information. Some of this information has been manually curated. The work some years in the making and the authors have used knowledge of mouse genetics to shed light on genetic influence for vertebral abnormalities in humans, maintaining power by selecting candidates a priori.

Major comments:

1. It would be helpful to have a title more reflective of the contents. There are many chronic pain GWAS reported from UK Biobank including those on back pain. I suggest the title be re-configured to contain the methodological details 'candidate gene' and perhaps 'whole exome sequencing' around vertebral anomalies, and back pain be removed.

Response: We thank the reviewer for this helpful suggestion. In line with this advice, we have revised the title to better reflect the methodological focus of the study as the following:

“Identification of novel vertebral development factors through UK Biobank candidate gene search and body imaging analysis”

2. Methods - there is a lack of detail around identifying the variants associated with (multiple)

phenotypes. What did the manual curation amount to? What methods were used to test association with variants? Please provide details

Response: We thank the reviewer for highlighting the need for additional methodological detail. Briefly, we began with a predefined panel of candidate genes relevant to vertebral development. Candidate variants were then filtered based on minor allele frequency ($MAF \leq 1\%$) and their association with sitting height. These associations are identified using publicly available PheWAS results from the AstraZeneca PheWAS Portal, which integrates whole-exome sequencing data from UK Biobank with ~10,000 binary and ~3,500 quantitative traits, including anthropometric measures.

In the revised manuscript, we have expanded the Methods section (lines 469-495) and updated the Supplementary Figure S5 to provide a clearer description of our variant identification and curation process:

“To identify specific candidate variants contributing to numerical variations in human vertebrae, we used AZPheWAS data and applied the following criteria: (1) restrict the analysis to 51 genes of interest (2,334 variants); (2) retain less common and rare alleles with a minor allele frequency ($MAF \leq 0.01$ within the UKB cohort (2,194 variants); (3) remove synonymous variants (1,499 variants); (4) identify variants associated with body size–related measurements within ICD-10 Chapter XXI (Factors influencing health status and contact with health services) at $P < 1 \times 10^{-5}$ to capture a broader range of potential candidates (13 variants; Table S4); and (5) further narrow to variants where sitting height showed the strongest association (4 variants), excluding those where associations were primarily driven by standing height, weight, body fat percentage, or trunk fat mass.”

3. What software was used to call the variants? Were CNVs considered as well as SNVs?

Response: We thank the reviewer for this comment. In this study, we did not perform variant calling ourselves, as this step had already been completed by the UK Biobank (Szustakowski *et al*). Four candidate single nucleotide variants (SNVs) of interest were identified (Methods section: Gene and Variant Selection), and carriers were ascertained using the UK Biobank’s centrally processed whole-exome sequencing (WES) dataset, the population-level exome OQFE variant dataset from the 450k release (Field ID: 23157; genome build GRCh38), which had undergone

variant calling and quality control by the UKB (Szustakowski et al., 2021). Analyses focused exclusively on these four SNVs; copy number variants (CNVs) were not considered. We used the UKB Research Analysis Platform (RAP) cohort browser to retrieve carriers of these variants directly.

We have updated the Methods section (lines 515-520) to clarify this point as follows:

“Carriers of the four candidate variants were identified using the UKB WES data, specifically the population-level exome OQFE variant dataset from the 450k release (Field ID: 23157; genome build GRCh38), provided by UKB. These data were accessed via the 'Genomics' section of the UKB Research Analysis Platform cohort browser (<https://ukbiobank.dnanexus.com/>). We retrieved a subgroup of 442 candidate variant carriers with body imaging data (detailed in the next section) available for vertebral and rib phenotyping (Table 3).”

Reference:

Szustakowski, J.D., Balasubramanian, S., Kvikstad, E. *et al.* Advancing human genetics research and drug discovery through exome sequencing of the UK Biobank. *Nat Genet* **53**, 942–948 (2021).

4. Multiple testing - how has this been taken into account? Particularly in the pain section, once multiple phenotypes have been considered, the level usual level of significance will likely not be met. Clarification would be appreciated.

Response: We thank the reviewer for highlighting this important point. We acknowledge the challenge of statistical testing across multiple categorical phenotypes, which we treated as categorical variables rather than converting to a continuous scale. To address multiple testing, we applied false discovery rate (FDR) correction for all phenotypic association tests. For the pain outcomes specifically, we performed FDR-adjusted one-sided tests to assess whether carriers reported increased pain compared to non-carriers. As noted, some associations did not remain significant after FDR adjustment, although with raw p values < 0.05 (figure S2).

We added this as a limitation in the Discussion (lines 406-410) as the following:

“A limitation of our study is that deriving vertebral and rib phenotypes was extremely labour-intensive, especially because we used non-ideal imaging data which required modal fusion to allow

analysis. This effort constrained the scalability of phenotyping, and although we analysed over 900 participants, statistical power remained limited for some downstream analyses, particularly secondary outcomes such as pain.”

Minor comments:

1. line 75 - please clarify how this WES study of candidate genes is proof of principle for GWAS - or remove the statement

Response: We thank the reviewer for pointing this out. We removed the paragraph in the Introduction and Discussion.

2. from 51 candidate genes 4 were selected containing variants. How was this done - simply by association with sitting height? Methods seem very sparse at this point, please expand.

Response: We thank the reviewer for this important comment. We agree that the original description was too brief, and we have now expanded the Methods section and revised Supplementary Figure S5 to provide additional details on the variant selection process. Sitting height was chosen as a key proxy phenotype because our aim was to identify variants associated with numeric anatomical anomalies (e.g., variation in vertebral number), rather than proportional differences in overall skeletal size (e.g., taller stature, longer limbs). We then applied frequency-based filtering (rare variants defined as $MAF \leq 1\%$) and assessed associations across body size measures.

To further clarify, we have added Supplementary Table S4, which lists all variants (both common and rare) from our candidate genes that showed significant associations with body size measurements. This table, together with the expanded methodological description and updated schematic in Supplementary Figure S5, provides a more transparent overview of how the four final variants were selected.

We expanded the Method (lines 469-495) details as the following:

“To identify specific candidate variants contributing to numerical variations in human vertebrae, we used AZPheWAS data and applied the following criteria: (1) restrict the analysis to 51 genes of interest (2,334 variants); (2) retain less common and rare alleles with a minor allele frequency

(MAF) ≤ 0.01 within the UKB cohort (2,194 variants); (3) remove synonymous variants (1,499 variants); (4) identify variants associated with body size–related measurements within ICD-10 Chapter XXI (Factors influencing health status and contact with health services) at $P < 1 \times 10^{-5}$ to capture a broader range of potential candidates (13 variants; Table S4); and (5) further narrow to variants where sitting height showed the strongest association (4 variants), excluding those where associations were primarily driven by standing height, weight, body fat percentage, or trunk fat mass.”

We further explained the rationale for these criteria as the following in the Methods section (lines 497-505):

‘We focused on rare variants (MAF $\leq 1\%$) because these are more likely to lead to congenital anomalies in vertebral development, such as numeric changes in vertebral or rib number. This also aligns with the reported prevalence of congenital numeric variations, such as lumbar and cervical ribs, which each occur in $<1\%$ of the population. Sitting height was chosen as a key proxy phenotype because our aim was to identify variants associated with numeric anatomical anomalies, such as variation in vertebral number. In contrast to other body measurement, sitting height is more directly influenced by the spine and therefore better suited to capture congenital differences in vertebral segmentation rather than proportional differences in overall skeletal size (e.g., taller stature or longer limbs).’

Typos

Line 91/2 should read 'numerical variation', Experiences of pain should read 'experience of pain'

Response: Thanks for pointing out these typos. We revised these as suggested.